# Optimizing Silage Strategies for Sustainable Livestock Feed: Preserving Retail Food Waste

Vicky Garcia Rodriguez [1], Layton Vandestroet [1], Vinura C. Abeysekara [1], Kim Ominski [2], Emmanuel W. Bumunang [1], Tim McAllister [3], Stephanie Terry [3], Luis Alberto Miranda-Romero [4] and Kim Stanford [1,*]

1   Department of Biological Sciences, University of Lethbridge, Lethbridge, AB T1K 3M4, Canada
2   Department of Animal Science, University of Manitoba, Winnipeg, MB R3T 2N2, Canada
3   Agriculture and Agri-Food Canada Lethbridge Research and Development Center, Lethbridge, AB T1J 4B1, Canada
4   Departamento de Zootecnia, Universidad Autónoma Chapingo, Texcoco 56230, Mexico
*   Correspondence: kim.stanford@uleth.ca

**Abstract:** In Canada, approximately 11.2 million metric tons of avoidable food waste (FW) is produced per year. Preservation of a greater proportion of this FW for use as livestock feed would have significant environmental and socioeconomic benefits. Therefore, this study blended discarded fruits, vegetables, and bakery products from grocery stores into silage to assess the ability to preserve their nutritional value and contribute to the feed supply. Two treatments for reducing the water content of FW were evaluated, sun-dried (SD) and passive-dried (PD), and compared to control (C) using laboratory mini-silos over 60 days of ensiling. Although dry matter (DM) was increased by 1–5% for PD and SD, respectively, up to 41.9% of bread products were required to produce a targeted silage DM of 38%. All mature silages were high in crude protein (15.2 to 15.7%), crude fat (6.0 to 6.3%), sodium (0.48 to 0.52%), and sugars (0.95 to 1.53%) and were low in neutral detergent fiber (6.2 to 7.6%) as compared to traditional silages used as livestock feed. Mold and other signs of spoilage were visible on FW, but mycophenolic acid was the only mycotoxin above the limit of detection in material prior to ensiling. Plate counts of molds and yeasts declined ($p < 0.001$) by 5–7 log colony-forming units (CFU) over 60 days of fermentation and were not detected in mature silage. All silages were aerobically stable over 20 days. This study indicates that FW can produce good-quality silage but approaches other than SD and PD are required for increasing silage DM as insufficient bread products may be available for this purpose in all batches of FW.

**Keywords:** silage; food waste; livestock feed; fruit; vegetables; bread

## 1. Introduction

Food waste is a serious problem that causes terrible environmental impacts in agriculture, such as the production of greenhouse gases [1], reduction of biodiversity, contamination of land and water with excess nutrients, herbicides and pesticides [2], and ecological degradation [3]. In Canada, 35.5 million metric tons of food is lost each year, of which 11.2 million metric tons is avoidable food waste [4]. Food waste in retail outlets is estimated to represent 10% of total waste, including fruits, vegetables, and cereals [2]. A substantial proportion of this waste comes from bread, accounting for up to 10% of the total food waste generated in several countries [5]. The urgency of addressing this problem aligns with the Sustainable Development Goals (SDGs) outlined by the United Nations, which emphasize a crucial reduction in food waste, especially at the consumer and retail levels [1]. Such measures aim to slow down the detrimental environmental effects associated with excessive food waste [6].

Efforts to mitigate food waste present multifaceted benefits, extending beyond environmental concerns to economic losses and nutritional waste [6,7]. Proponents have

emerged to comprehensively address these challenges for reusing food waste as animal feed [3]. This approach provides a promising avenue for recovering the nutritional value of food that would otherwise go to waste, reducing not only the environmental burden but also economic losses associated with waste disposal [1], although some food waste is unavoidable [8].

A factor that precludes the use of waste fruits and vegetables is their high water content (76–93.5%), which contributes to rapid spoilage [3]. One option to reduce spoilage and preserve waste fruits and vegetables is to ensile them [9]. Ensiling is a controlled microbial fermentation process [10] traditionally used to preserve forages for livestock feed. It is a relatively easy and low-cost preservation method through two main mechanisms: (A) establishment of an anaerobic environment and (B) fermentation of plant sugars to lactic and other organic acids, which lowers pH and prevents the growth of spoilage microorganisms including molds, yeasts, and bacteria [11].

Previous studies of ensiling waste vegetables, fruits, citrus, and bakery products alone or mixed with other forage crops to produce feed suitable for livestock are limited [2]. Some of these food waste silages have been shown to have a nutritional value comparable to good-quality forages commonly fed to dairy cattle [12]. Most of these studies have focused on waste vegetables as they can serve as sources of crude protein and micro- and macro-minerals and usually have sufficient water-soluble carbohydrates to support silage fermentation [12,13]. Surplus bread, historically repurposed as animal feed, underscores the potential of diverting food waste towards livestock. However, key concerns revolve around moisture content and nutritional variability, necessitating careful waste separation to prevent contamination [5]. Recent studies focus on the efforts to effectively utilize surplus bread as part of diets for livestock could mitigate nutritional variations. Other studies have focused on bread waste and assessed its optimal level of inclusion in the diets of pigs, poultry, and rabbits [7].

Extensive research is essential to fill this knowledge gap by elucidating effective ensiling methodologies for various food waste streams and their feasibility for sustainable livestock feeding. Therefore, the objectives of this study were to (1) assess the feasibility of producing silage solely from retail food waste; and (2) assess the ability of two ensiling practices to generate moisture contents that would produce high-quality FW silage for ruminant livestock.

## 2. Materials and Methods

### 2.1. Food Waste Processing

Food waste (FW), consisting of fruits, vegetables, bread, and bakery products (see below), was obtained from a local grocery store from their daily stock slated for disposal after nearing or reaching its expiration date. Plastic or other packaging was manually removed from the FW before chopping to a length of 1 cm$^2$ using a CL50 continuous feed food processor (Robot-Coupe USA Inc., Ridgeland, MS, USA). Bread and bakery products were allowed to air dry in a cool dry environment for 24 h before chopping similarly to fruits and vegetables and were stored separately from fruits and vegetables. Chopped fruits and vegetables were divided into three equal portions. Control (C) was kept refrigerated in sealed bins, whereas the other two treatments were placed in layers 5–10 mm deep on plastic screens suspended on metal frames to drain. One treatment was exposed to solar radiation to be dried (SD), while the other was left inside the building to be dried passively (PD). The study was conducted on days with an average ambient temperature ranging from 9 to 14 °C and relative humidity from 76 to 85%. Drying treatments had a duration of 12 h. The DM content of each treatment and processed air-dried bread and bakery products was assessed using a microwave oven [14]. The fruits and vegetables from the three treatments were then thoroughly mixed with the processed air-dried bread and bakery products. Proportions averaged 59.5% (wet weight) fruits and vegetables and 40.5% bread and bakery products to obtain a target ~38% silage DM (range 37.0 to 38.2%). The vegetables used and listed in order of abundance included potatoes, beets,

onions, cabbage, carrots, cucumbers, radishes, squash, lettuce, hot peppers, bell peppers, mushrooms, radicchio, bok choy, celery, mixed greens, parsnips, turnips, eggplants, and okra. Fruits included apples, pears, strawberries, grapes, oranges, mangoes, pineapples, star fruits, and tomatoes. Bakery products included bread, crackers, cookies, muffins, English muffins, buns, bagels, and garlic and plain naan bread. Trace amounts of fresh spices such as mint, rosemary, garlic, parsley, basil, cilantro, and ginger were also present in the FW and were mixed with fruits and vegetables.

## 2.2. Preparation of Mini-Silos

Five mini-silos of 10.3 cm in diameter × 35.5 cm in height [15], and with an average weight of 1.33 kg, were prepared per FW drying treatment. The mini-silos were filled and compacted using a hydraulic press to achieve a target density of 240 kg/m³. Representative samples of the ensiled mixture for each treatment were collected in triplicate for pH, mycotoxins, and enumeration of microbial populations. Each mini-silo was weighed before and after filling, sealed, and stored at room temperature (~20 °C). Silos were opened after 60 d of ensiling and weighed to calculate the DM loss based on the initial weight of each silo on D0. On D60, representative samples of the three treatments of FW silage were collected in triplicate for analyses of pH, feed quality, volatile fatty acids (VFA), lactic acid, ammonia, mycotoxins, and enumeration of microbial populations.

## 2.3. Microbial Analyses: Lactobacilli, Yeasts, Molds, and Mycotoxins

Each FW silage sample (10 g) was added to 90 mL of sterile 70 mM potassium phosphate buffer (pH 7.0) and stirred using a magnetic stirrer for 5 min. The suspension was then serially diluted from $10^2$ to $10^7$ in triplicate and plated onto DeMan Rogosa, and Sharpe (MRS, Dalynn, Calgary, AB, Canada) semi-selective medium, supplemented with 200 μg/mL cycloheximide for enumeration of *Lactobacillus* spp. after incubation at 37 °C for 24–48 h. Determination of yeasts and mold used triplicate plates of Sabouraud dextrose agar (SDA, Dalynn) amended with 100 μg/mL each of tetracycline and chloramphenicol. Yeast and mold colonies were enumerated after plates were allowed to stand at ambient temperature for 72 h. In all cases, colonies were counted from plates, with a minimum of 30 and a maximum of 300 colonies [10]. Mycotoxin analyses were conducted by a commercial laboratory (Actlabs, Ancaster, ON, Canada) for a panel of 16 common mycotoxins (aflatoxins B1, B2, G1 and G2, deoxynivalenol, 3-acetyl-deoxynivalenol, 15-acetyl-deoxynivalenol, fumonisins B1 and B2, ochratoxin A, T-2, HT-2, zearalenone, diacetoxyscirpenol, sterigmatocystin, and mycophenolic acid) using liquid chromatography and tandem mass spectroscopy instrumentation [16].

## 2.4. Silage Aerobic Exposure

After 60 d of ensiling, all FW silos were opened to air and approximately 100 g per silo was taken for other analyses. To study the stability upon aerobic exposure of silage [17], the remaining content of silage in each silo was mixed thoroughly in the silo and covered with two layers of cheesecloth to prevent drying and contamination but allowing penetration of air at room temperature for 20 d. A Traceable® Snap–in module thermometer (VWR International, Edmonton, AB, Canada) with a probe was used to measure the temperature 10 cm from the open face of the silage daily and compared to room temperature. The contents of each mini-silo were subsampled after 3, 7, 14, and 20 d of aerobic exposure for pH determination. Aerobic stability was calculated as the number of hours that silage was exposed to air before the temperature of aerobically exposed silage exceeded the baseline ambient temperature by 2 °C [18].

## 2.5. Chemical Analyses

Triplicate samples (15 g) from each treatment of FW silage before (D0), after the silage fermentation process (D60), and after aerobic exposure were mixed with 135 mL of deionized water and blended at full speed for 30 sec using a blender (Waring Commercial,

Torrington, CT, USA). The suspension was filtered through two layers of cheesecloth, and the pH was determined using an Accumet AB150 pH meter (Fisher Scientific, Ottawa, ON, Canada). The filtrate (7.5 mL) was immediately boiled for 10 min to stop fermentation and stored at $-20\,°C$ until analyzed for water-soluble carbohydrates using the Nelson–Somogyi method [19]. Water-soluble carbohydrates were expressed as glucose equivalents by reducing the cupric form of $Cu^{2+}$ to the cuprous $Cu^{+}$ and reading absorbance at 620 nm [15]. The remainder of the filtrate was centrifuged at $10,000\times g$ for 15 min at $4\,°C$. The supernatant was collected and frozen at $-40\,°C$ for subsequent analyses of volatile fatty acids (VFA), lactic acid, and ammonia, as described by Nair et al. [10].

### 2.6. FW Nutritional Quality

Duplicate samples (150 g) of mature silage were freeze-dried and ground to pass through a 1 mm screen. Samples were analyzed by a commercial feed-testing laboratory (Down to Earth Labs Inc., Lethbridge, AB, Canada). Crude fat was determined by ether extraction according to AOAC method 920.29; ash was determined according to AOAC method 942.05 after heating 1 g of silage at $600\,°C$ for 2 h. Minerals were determined using atomic absorption according to AOAC method 935.13A [20]. Crude protein was determined colorimetrically using flow injection analysis according to AOAC method 984.13 [20]. The acid detergent fiber (ADF) was determined by adding 0.5 g samples of silage to filter bags and digesting for 75 min in 2 L of ADF solution in an Ankom Delta digestion unit (Ankom Technology, Macedon, NY, USA). An Ankom digester was also used to determine neutral detergent fiber (NDF) according to the methods of Van Soest et al. [21].

### 2.7. Statistical Analysis

All statistical analyses were conducted using SAS (SAS 9.4; SAS Institute Inc., Cary, NC, USA). Nutritional and fermentation data were analyzed using mixed models (PROC MIXED) in a completely randomized design, with drying treatments (C, PD, and SD) as fixed effects. For the microbial analysis, data were analyzed using generalized linear mixed models (PROC GLIMMIX) in a binomial distribution. In all cases, differences among individual least square means were considered significant at $p < 0.05$.

## 3. Results and Discussion

### 3.1. FW Characteristics

Food waste was obtained from the grocery store in the summer season (June 2022). Physical evaluation of the quality of the FW was performed, with fruits and vegetables that presented excessive mold growth or complete deterioration being discarded. All FW was within the general categories of fruits, vegetables, bread and bakery products, and spices (see Section 2.1). Generally, the chopping and handling of the FW was rapid and uncomplicated. It was accompanied by the release of water from most fruits and vegetables. Some fruits such as strawberries, tomatoes, limes, and oranges, and vegetables such as cucumbers, released a large amount of water. The ready-to-eat salads consisted of lettuce, spinach, shredded carrots, broccoli, and shredded cabbage and, as already chopped, were easy to process. Bread and bakery products were left in a dry and well-ventilated area prior to processing, which allowed them to lose moisture and facilitated their processing. Some bakery products like cream-filled cakes were not included as they were too soft to be chopped in the food processor and potentially included dairy products. Dairy and meat products were removed from the waste stream to comply with the restrictions of not feeding ruminant proteins to ruminants [22]. Bakery products constituted 38.7% to 41.9% of silage (Table 1) and enabled a target DM of 38% in processed and blended FW with no leachate formation. Avoiding leachate would be an important consideration for increasing the scale of FW silage production, as it can contaminate soil and result in a loss of nutrients from the silage [23].

**Table 1.** Fermentation characteristics of ensiled FW passively dried (PD) and sun-dried (SD) for 12 h and refrigerated FW control (C).

| Items | Treatments | | | SEM | *p*-Value |
|---|---|---|---|---|---|
| | **C** | **PD** | **SD** | | |
| FV * weight D0, % | 58.95 [a] | 58.32 [a] | 61.31 [b] | 0.21 | <0.0001 |
| BBP * weight D0, % | 41.04 [a] | 41.67 [a] | 38.7 [b] | 0.32 | <0.0001 |
| DM FV D0, % | 10.47 [a] | 11.59 [a] | 15.54 [b] | 1.15 | 0.047 |
| Initial DM of FW * (D0), % | 36.97 [a] | 38.21 [b] | 36.88 [a] | 0.19 | 0.001 |
| Final FW DM (D60), % | 35.54 [a] | 38.07 [c] | 36.82 [b] | 0.21 | <0.0001 |
| Final weight loss (D60), % | 3.93 | 2.92 | 3.09 | 0.60 | 0.467 |
| Initial pH | 5.26 [ab] | 5.35 [b] | 5.10 [a] | 0.05 | 0.032 |
| Final pH | 3.85 [a] | 3.86 [a] | 4.03 [b] | 0.02 | 0.002 |
| Volatile fatty acids, % of FW silage DM | | | | | |
| Acetic acid | 2.16 [a] | 1.75 [b] | 2.22 [a] | 0.33 | 0.0001 |
| Propionic acid | 0.115 [a] | 0.114 [a] | 0.100 [b] | 0.001 | 0.002 |
| Butyric acid | 0.013 | 0.009 | 0.010 | 0.001 | 0.10 |
| Organic acid concentration, % of silage DM | | | | | |
| Lactic acid | 7.50 [a] | 8.40 [b] | 7.51 [a] | 0.10 | 0.001 |
| Succinic acid | 0.083 | 0.095 | 0.866 | 0.001 | 0.003 |
| $NH_3$, % of crude protein | 4.96 [a] | 4.90 [a] | 4.15 [b] | 0.03 | <0.0001 |
| Microbial analysis, $\log_{10}$ CFU $g^{-1}$ DM | | | | | |
| *Lactobacillus* spp., D0 | 7.2 [a] | 7.8 [b] | 7.7 [b] | 0.06 | <0.0001 |
| *Lactobacillus* spp. D60 | 5.1 [a] | 5.3 [b] | 5.6 [c] | 0.06 | <0.0001 |
| Molds and yeasts, D0 | 5.6 [a] | 7.2 [c] | 6.6 [b] | 0.02 | <0.0001 |
| Molds and yeasts, D60 | nd | nd | nd | nd | nd |

Values with different letters indicate differences between means. DM, dry matter; SEM, pooled standard error of mean; FV, fruit and vegetables; BBP, bread and bakery products; FW, food waste—see text (Section 2.1) for further details; D0, initial ingredients before ensiling; D60, after 60 days of ensiling; nd, not detected. * The mean dry matter of bread and bakery products (75%) was used to estimate silage DM at D0. Means with different superscripts differ (*p* < 0.05).

Many FW products had plastic packaging, such as ready-to-eat salads and fruits in net bags, with most bakery products also in plastic packaging. In total, plastic packaging accounted for 3.5% of the weight of FW gathered per load, which was removed manually and slowed silage preparation. To scale up silage production from FW, a mechanized and effective means of removing plastic packaging would be required, as plastic would interfere with the ensiling process and be detrimental to livestock health. Large pieces of plastic in the feed may lead to blockage of the gastrointestinal tract, internal injuries, and potentially livestock death [24]. In contrast, microplastics may bioaccumulate and negatively impact livestock health, growth, or reproduction as well as being potentially transferred to humans consuming the animal products [25].

### 3.2. Fermentation Characteristics

Fermentation parameters for FW silage were generally similar to those of conventional cereal silage for livestock and even though differences were found among the three drying treatments, all were within expected ranges (Table 1). Weight of FW at initial ensiling averaged 3.19 kg and 3.09 kg after 60 d ensiling. Accordingly, dry matter loss after 60 d fermentation ranged from 2.92% to 3.93%, similar to results reported by Nair et al. [26] for well-preserved barley silage. In the present study, the pH of FW silage dropped from a maximum of 5.26 to a minimum of 3.85 after 60 d of ensiling, as has been reported for high-quality corn silage [23]. Lactic acid production for the three treatments was similar to that reported for grass silage [12,23]. Silage with DM between 30 and 40% reduces the risk of heat damage and increases lactic acid production during ensiling [27]. Butyric acid production

was low, suggesting that the growth of spoilage bacteria like *Clostridium* spp. was inhibited by the low pH [28]. Concentrations of ammonia in the present study were consistent with those detected by Dou et al. [12] for fresh fruit and vegetable silage, indicating that protein loss after the ensiling process was relatively low, and suggesting minimal degradation of proteins in silage [29]. Although the incorporation of essential oils from fresh herbs and spices has been shown to improve fermentation and aid lactic acid bacteria growth in some cases, essential oils have also inhibited fermentation and aerobic stability of corn silage [27]; fresh herbs and spices were only present in trace amounts in the FW evaluated and their possible impacts were likely diluted by other FW constituents.

### 3.3. Microbial Analysis

A diverse microbial population is normal in fresh fruit and vegetable tissues and in traditional cereal silages [30,31]. Fermentation under anaerobic conditions was quickly established, which was reflected in the decrease in pH, production of organic acids and VFA, and inhibition of molds and yeasts (Table 2), indicating that proper anaerobic fermentation had been achieved [27]. Populations of *Lactobacillus* spp. were similar among the three treatments after 60 d fermentation, and their decrease from the moment of ensiling (D0) to the end of the fermentation (D60) indicates that the epiphytic bacterial community was influenced by fermentation during ensiling [28].

**Table 2.** Feed quality of ensiled FW treatments: passively dried (PD), sun-dried (SD), or refrigerated control (C).

| Item | Treatment | | | SEM | *p*-Value |
|---|---|---|---|---|---|
| | C | PD | SD | | |
| Chemical composition, % DM | | | | | |
| Crude protein | 15.4 | 15.2 | 15.7 | 0.49 | 0.82 |
| ADIN | 0.49 | 0.47 | 0.61 | 0.23 | 0.91 |
| ADF | 4.56 [a] | 4.79 [a] | 6.32 [b] | 0.14 | 0.01 |
| NDF | 6.21 | 6.16 | 7.63 | 0.32 | 0.08 |
| WSC | 1.20 [a] | 0.95 [a] | 1.53 [b] | 0.08 | <0.0001 |
| Crude fat | 6.02 | 6.28 | 5.97 | 2.44 | 1.0 |
| Ash | 4.56 [a] | 4.79 [a] | 6.32 [b] | 0.14 | 0.01 |
| Macrominerals, % DM | | | | | |
| Potassium | 0.92 [a] | 0.96 [b] | 1.12c | 0.02 | 0.02 |
| Phosphorous | 0.250 [a] | 0.245 [a] | 0.270 [b] | 0.002 | 0.02 |
| Calcium | 0.190 [a] | 0.195 [a] | 0.230 [b] | 0.002 | 0.01 |
| Magnesium | 0.10 | 0.10 | 0.11 | 0.01 | 0.44 |
| Sodium | 0.52 | 0.50 | 0.48 | 0.01 | 0.26 |
| Sulfur | 0.14 | 0.18 | 0.16 | 0.02 | 0.45 |
| Trace elements, mg/kg DM | | | | | |
| Copper | 9.5 | 5.4 | 7.7 | 2.4 | 0.54 |
| Manganese | 18.1 | 17.7 | 19.4 | 0.40 | 0.10 |
| Zinc | 25.3 | 25.0 | 25.4 | 1.0 | 0.94 |
| Iron | 97.5 [ab] | 82.2 [a] | 107.5 [b] | 3.5 | 0.03 |

FW, food waste—see text (Section 2.1) for further details; ADIN, acid detergent insoluble nitrogen; ADF, acid detergent fiber; NDF, neutral detergent fiber; WSC, water-soluble carbohydrate; DM, dry matter; SEM, standard error of mean. Means within a row with different letter superscripts differ significantly ($p < 0.05$). Starch and pectins not included in analyses.

In this trial, some of the fruits and vegetables that were used as silage raw materials had visible mold prior to ensiling, which explains the relatively high (5–7 $\log_{10}$ CFU) initial counts of mold and yeast (Table 1). During ensiling, there was approximately a 2 log CFU decrease in *Lactobacillus* spp., with 5.1 to 5.6 CFU/g DM present in silage at D60. In contrast to *Lactobacillus* spp., the growth of molds and yeast was completely inhibited in

D60 samples. The typical population of lactic acid bacteria and fungi including yeasts and molds on plants before ensiling ranges from 5 to 9 $\log_{10}$ CFU/g of crop [32]. The decrease in the *Lactobacillus* spp. noted during ensiling in the present study was probably due to the reduced pH of the mature silage. In addition, under aerobic conditions, as would be present at D0, lactic acid bacteria commonly produce hydrogen peroxide, which can inhibit the same bacteria [23].

*3.4. Mycotoxins*

As molds were observed on some fruits and vegetables during processing, there was a concern that mycotoxins may be introduced into FW silage.

However, almost all mycotoxins assayed were below the corresponding limits of detection (LOD) at D0 and in D60 silage. The LOD were 1 µg/kg for aflatoxins B1, B2, G1, and G2; 0.06 mg/kg for deoxynivalenol, 3-acetyl-deoxynivalenol, 15-acetyl-deoxynivalenol, T-2, HT-2, and diacetoxyscirpenol; 0.1 mg/kg for fumonisins B1 and B2; 3 µg/kg for ochratoxin A; and 0.03 mg/kg for zearalenone, sterigmatocystin, and mycophenolic acid. Only one mycotoxin, mycophenolic acid, was detected at a concentration of 0.07 mg/kg in PD and SD samples at the beginning of the trial (D0), whereas the concentration of this mycotoxin was below the LOD in the C samples at D0 and in all treatments (C, PD, and SD) after 60 days of ensiling. Mycophenolic acid is known to have immunosuppressive properties [33] and has been detected in a survey of silages at an average concentration of 4.2 ppm [34]. As concentrations of mycophenolic acid in D0 FW silage were at least 20 times lower than that detected in grass silage by Manni et al. [35] and similar to that detected in forage crops before ensiling [36], its presence in this FW silage is unlikely to pose a risk to livestock health. The results of this study indicate that the occurrence of mycotoxins in FW was adequately controlled by fermentation. Future evaluations of FW silage for mycophenolic acid and other mycotoxins would be advisable, as mold concentrations in FW vary and can increase if oxygen is not rapidly removed and anaerobic conditions are not maintained during ensiling [36]. Additionally, the fruits, vegetables, bread, and bakery products in the present study were ready for sale in grocery stores. Consequently, these FW sources may be less susceptible to contamination by spoilage microorganisms than field-harvested forages that are directly ensiled [3].

*3.5. Nutritional Quality of the FW Silage*

The feed quality of the FW drying treatments is presented in Table 2. Crude protein (CP) of FW silage treatments ranged from 15.25% to 15.70%, similar to levels in typical alfalfa hay and grass hay silages [12], slightly higher than that of barley silage [26], and approximately double that of corn silage [37]. Both acid detergent fiber (ADF) and neutral detergent fiber (NDF) were lower than those found in traditional silage [10]. Although Dou et al. [12] reported 13.6% ADF and 15.9% NDF for ensiled fruits and vegetables, fiber concentrations of silage in the present study were even lower than those, likely due to the inclusion of bread and bakery products. Typically, the average fiber content of bread and bakery products ranges from 1.6 to 6.5% [7].

The crude fat value of the FW silage ranged from 5.97 to 6.28%, higher than that of cereal and corn silage [37], but lower than that found by Froetschel, et al. [2], which averaged 11.6% for an ensiled mixture of fruit, vegetables, and bakery products seeped for 24 h. Feeding high-oil corn silage with crude fat concentrations similar to our FW silage increased the yield of fat-corrected milk by dairy cattle as compared to traditional corn silage (3.4% crude fat) [37]. Similar to crude fat, sodium concentrations of FW silage were higher than is typical for traditional silage. Barley silage averages 400 ppm sodium [38], with FW silage having approximately 12 times more sodium, likely arising from the presence of bread and bakery products. Although sodium chloride is commonly used to preserve food by inhibiting bacterial growth, lactic acid bacteria are relatively salt tolerant, while spoilage organisms such as *Clostridium butyricum* are more sensitive [39].

Consequently, the high salt content of the FW silage may have also inhibited the growth of some spoilage microorganisms.

Water-soluble carbohydrate (WSC) concentrations were similar to those reported by Froetschel et al. [2] for ensiled FW and were approximately double that of conventional barley silage [40]. Concentrations of glucose and fructose would be expected to be high with the inclusion of fruits and bakery products in silage, with sugars subject to rapid fermentation to volatile fatty acids, resulting in a rapid drop in pH [41]. Ash concentrations were 4.56% for C, 4.79% for PD, and 6.32% for SD FW silage, within the range of values obtained by Forwood et al. [9] for corn ensiled with 40% of either carrots or pumpkins. Although significant differences were present across drying treatments for ADF, sugars, calcium, and iron, all results were within the ranges previously reported for silage [9,41,42]. As the method of drying would be unlikely to alter concentrations of sugars, fiber, or minerals, differences in nutrient composition across treatments noted in feed quality parameters likely reflect the variability in FW included in each treatment; i.e., if more spinach was included in SD, iron concentrations would be expected to be elevated. Of more concern for drying treatments were the concentrations of ADIN, which provides an estimate of heat-damaged proteins [26], although neither drying treatment differed in ADIN from control.

*3.6. Silage Aerobic Stability*

The surface temperature of FW varied depending on the drying treatment; however, in all treatments, the surface temperature did not exceed 2 °C above ambient temperature, demonstrating stability of the silage over 20 d aerobic exposure. Similarly, pH did not increase to more than 4.02 in any treatment over 20 d of aerobic exposure. Although silage quality in mini-silos should be verified on-farm in larger-scale silage, these aerobic stability results are promising. Aerobic stability demonstrated that nutrients were preserved, and minimal amounts of mold spores were present, with 7 d of aerobic stability a target for on-farm silage [43]. The creation of a stable FW silage is important as a stable silage would allow efficient long-term use of all FW available. Further, the creation of a stable supply of feed is important because it would allow for testing and nutritional analysis before feeding, allowing the diet to be optimized to better meet livestock requirements.

*3.7. Challenges and Practical Considerations for Converting FW to Silage for Livestock Feed*

Previous studies have evaluated by-products of the agri-food industry as total or partial ingredients of silage. Hooker et al. [44] used 60% forage corn with unsalable vegetables to produce high-quality livestock feed. Valdez-Arjona et al. [45] replaced up to 77% of corn silage with residues of *Curcubita argyrosperma*, indicating that it is feasible to use *C. argyrospema* as an alternative feedstuff for cattle in a diet consisting of silage, grazed pastures, molasses, and bran. Forwood et al. [46] concluded that feeding unsaleable carrots at 45% DM in a TMR can improve lamb performance and carcass characteristics, maintaining meat quality. Abo-Donia et al. [47] made two partial substitutions of grain by dates (up to 75%) as a source of energy for a corn stalk silage. These researchers indicated that partial replacement of yellow corn by discarded dates as a source of energy in silage did not show any negative effect on digestibility, feed efficiency, milk yield, and composition in crossbred Friesian cows. Additionally, orange pulp and sweet potato roots have been evaluated as silage ingredients for growing and finishing pigs, demonstrating that introducing silage in basal diets would contribute to an increase in fiber levels, with appreciable digestible energy content and potentially reduce ammonia and methane emissions [48,49].

The nutritional value of bread fed directly without ensiling to beef steers and pigs has also been studied [5,50]. However, there is limited information on silage from mixtures of discarded fruits, vegetables, and bakery products [2,12]. By-products from food processing, such as orange pulp, offer the advantage of a more consistent composition and volume compared to the variable FW stream from grocery stores. The composition of FW from

grocery stores is unpredictable and includes cardboard and plastic packaging, which must be removed before ensiling.

The volume of food waste in Canada accounts for 7 million tons per year in the pre-harvest, post-harvest, processing, and distribution stages [1]. Nevertheless, the availability of grocery FW on any given day is unpredictable, with the surplus beyond what can be rapidly fed to livestock being the primary target for silage production.

The drying techniques evaluated were chosen for their low cost and relative ease of use but were largely ineffectual as SD increased the DM of FW by 5% compared to control, with PD only showing a 1% increase in DM. Other mechanized or more labor-intensive, but more effective, drying methods for food waste have been developed [51], but these may significantly increase the cost of utilizing food waste as livestock feed. In the present study, bread and other bakery products, such as cookies and crackers, were incorporated to increase the DM of the FW mixture for optimal fermentation and were in abundant supply for the batches of FW silage produced. However, due to potential shortages of these products during ensiling, other readily available high-DM by-products such as hay, straw, or cereal hulls may need evaluation as constituents to achieve desirable moisture levels for ensiling.

Converting FW to silage has significant implications for increasing nutrient availability in feed, particularly for non-ruminants such as pigs. Ensiling breaks down proteins and enhances the digestibility of starch [52]. The production of VFAs is crucial, as both ruminants and monogastric animals utilize VFAs as a source of energy [53]. Accordingly, FW silage in the present study would be a suitable feed for pigs [54], although considerations such as low DM content of silage may limit its use in swine operations employing dry-feed systems. Considering that dietary needs vary among livestock, beef cattle require higher ADF and NDF concentrations than would non-ruminants. For example, dairy cows typically need around 30% NDF and approximately 20% ADF to support rumen function, with variations based on lactation stage [55]. For this reason, FW silage, with its low ADF and NDF concentrations, would be suitable only as part of total mixed rations for beef or dairy cattle.

## 4. Conclusions

This study demonstrated that it is possible to produce high-quality silage solely from retail FW and suggests that this FW silage may serve as a valuable source of energy, protein, and minerals that can be a part of the diet of beef or dairy cattle and pigs. However, the low-cost drying techniques assessed for FW were of limited value. As the fermentation of silage in mini-silos does not always accurately represent larger-scale silage production, future evaluations of FW mixtures under on-farm ensiling conditions are necessary. In addition, larger-scale silage trials will allow animal feeding trials considering factors such as palatability of FW silage. Unfamiliar flavors such as those from hot peppers and spices, and variations in texture and composition, may lead to feed refusals. It is important to underline that to have high-quality FW from retail grocers suitable for ensiling as feed, it is vital to process it rapidly, especially during the warm season. Ideally, this would be within one to two days after collection. Otherwise, excessive mold growth can occur, as has been seen in other studies. In addition, prolonged storage prior to processing would contribute to further softening of fruits and vegetables, which complicates the processing operation. This study demonstrated that FW mixtures may yield high levels of lactic acid and inhibit spoilage microorganisms during ensiling. It is crucial to note that the conversion of FW into silage preserves its nutritional qualities and would remove the need for immediate feeding of FW from retail grocers. Additionally, preservation of FW as silage would allow for assessment of nutrient composition of silage prior to feeding, an important consideration due to the variable composition of retail FW.

**Author Contributions:** Conceptualization, K.S., K.O. and T.M.; methodology, K.S. and K.O.; formal analysis, K.S.; investigation, V.G.R., L.V. and V.C.A.; data curation, K.S. and V.G.R.; writing—original draft preparation, V.G.R.; writing—review and editing, V.G.R., L.V., V.C.A., E.W.B., T.M., S.T., L.A.M.-R. and K.S.; funding acquisition, K.S., K.O., T.M. and S.T. All authors have read and agreed to the published version of the manuscript.

**Funding:** The funding received from Secretaría de Educación, Ciencia, Tecnología e Innovación (fellowship: SECTEI/155/2021, Vicky García Rodriguez) and The Canada Food Waste Challenge of Agriculture and Agri-Food Canada is gratefully acknowledged.

**Data Availability Statement:** The data presented in this study are available on request from the corresponding author.

**Acknowledgments:** Many thanks to Loop Resources Canada for sharing their food waste. Many thanks to Hee-Eun Yang and D. Vedres for their technical support on this project.

**Conflicts of Interest:** The authors confirm that there are no known conflicts of interest associated with this publication and that there is no significant financial support for this work that could have influenced its outcome.

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
