# Peer review of "Optimizing Silage Strategies for Sustainable Livestock Feed: Preserving Retail Food Waste"

_agriculture, doi:10.3390/agriculture14010122_

Round 1
Reviewer 1 Report
Comments and Suggestions for Authors
Dear editor and authors,
This paper introduces a good reference for ensiling local FW as animal feed. It's positve for circular bio-enconomy and interesting for readers. Some errors should get revision before acceptance.
1 Title: this title is too broad, you should better refine your keywords. A optimized strategy ensiling retail food waste for livestock feed: restraining hazardous microflora and ····et al. You can refer to Paper1 https://doi.org/10.1016/j.eti.2022.102937
2 introduction: this part is not enough. You should introduce the background from 3~4 aspects. What: the retail food waste;Why: FW could be ensiled as feed; Who: other peers reported related research (should combine with what your highlight results, and emphasize your knowledge-gap). final, what's your experiment scheme. Refer to Paper2 https://www.mdpi.com/journal/microorganisms
3 Section 2.2: '×' rather than 'X'
4 Section 2.3: DeMan Rogosa, and Sharpe
5 Section 2.5.1: Cu2+?
6 Section 2.7: why some subtitles use capital letter? and some not?
7 Section 3.1: Table 1 should be put below the first arising.
8 References: Too many old references. Update them in 10 years except methods.
9 Another query: Why use bread waste to balance moisture? Generally, we use crop straws because of a big amount. Do you mix the raw materials refering to local food waste ratio? Bread wastes also have a big amount? These questions should be discussed at Introduction.
Author Response
This paper introduces a good reference for ensiling local FW as animal feed. It's positive for circular bio-enconomy and interesting for readers. Some errors should get revision before acceptance.
- Title: this title is too broad, you should better refine your A optimized strategy ensiling retail food waste for livestock feed: restraining hazardous microflora and····························· et al. You can refer to
Paper1 https://doi.org/10.1016/j.eti.2022.102937
Answer: Thank you for your valuable suggestion. We have changed the title, aiming to make it more specific, based on the suggested reference.
- introduction: this part is not You should introduce the background from 3~4 aspects. What: the retail food waste;Why: FW could be ensiled as feed; Who: other peers reported related research (should combine with what your highlight results, and emphasize your knowledge-gap). final, what's your experiment scheme. Refer to Paper2 https://www.mdpi.com/journal/microorganisms
Answer: Thank you for your thoughtful review of the introduction section of our manuscript. We appreciate your valuable comments and have carefully considered your suggestions for improving the clarity and completeness of the introduction. In response to your comments, we revised the introduction to provide a more comprehensive background from multiple aspects, incorporating the What ,Why and Who - In addition, we have checked the content of Paper2 of the journal Microorganisms (https://www.mdpi.com/journal/microorganisms). This helped us to align our content with that of the journal to improve and strengthen our introduction.
Answer: Section 2.2: '×' rather than 'X'
Answer: This has been corrected as suggested.
- Section 3: DeMan Rogosa, and Sharpe
Answer: Thank you for catching the error in the spelling of 'DeMan Rogosa, and Sharpe' in Section 2.3. Your correction is noted and will be implemented.
- Section 5.1: Cu2+?
Answer: Thank you for pointing out the correction regarding the notation for copper ions. I will use 'Cu2+' instead of 'Cu++' in the manuscript.
- Section 7: why some subtitles use capital letter? and some not?
Answer: Thank you for bringing attention to the use of capital letters in Section 2.7. After reviewing the subtitles, I realized the inconsistency. I believe you are referring to 'Chemical Analysis' and 'Statistical Analysis.' I will make the necessary adjustments by changing it to 'Statistical Analysis' and ensuring consistency, such as using lowercase for 'Results and Discussion
- Section 3.1: Table 1 should be put below the first arising.
Table 1 has been moved and is now located at the end of the paragraph after it was first mentioned.
- References: Too many old references. Update them in 10 years except methods.
This has been done and updated references are highlighted in revised version.
- Another query: Why use bread waste to balance moisture? Generally, we use crop straws because of a big Do you mix the raw materials refering to local food waste ratio? Bread wastes also have a big amount? These questions should be discussed at Introduction.
Answer: We wanted to assess the feasibility of making silage totally from retail food waste. There are substantial volumes of bread waste available in the food waste. In our batches of silage there was always sufficient bread to mix in. Additional information about bread and its availability added to introduction.
Reviewer 2 Report
Comments and Suggestions for Authors
The reviewed article raises very important issues regarding the reuse of food industry residues. Very often they are treated as waste, while they can be a raw material that can be reused. The aim of the reviewed work was to assess the ability of two ensiling practices to generate moisture con-tents that would produce high-quality FW silage for ruminant livestock. The methods adopted in this work allowed for the characterization of the analyzed substrates and their suitability for the ensiling process and, consequently, for becoming food for farm animals. The research results were presented in a transparent way against the background of available literature. At the end of the work, conclusions are presented, which rightly note certain limitations in translating laboratory research into real conditions. Nevertheless, I consider the work to be a valuable contribution to issues related to the circular economy. I only have a few comments that should be taken into account before publication:
1. There is no need to separate subsection 2.5.1 if it is the only one in section 2.5.
2. Please specify how much the mini silos weighed?
3. The citation of (CFIA, 2015) in section 3.1 is inappropriate. It should be [21].
4. In chapter 3.1 there is no description of the studied food wastes (FW). I suggest expanding this chapter.
5. The authors did not mention anything about how quickly food waste should be ensiled so that it can be used as food for animals.
Author Response
Reviewer #2
Comments and Suggestions for Authors
The reviewed article raises very important issues regarding the reuse of food industry residues. Very often they are treated as waste, while they can be a raw material that can be reused. The aim of the reviewed work was to assess the ability of two ensiling practices to generate moisture contents that would produce high-quality FW silage for ruminant livestock. The methods adopted in this work allowed for the characterization of the analyzed substrates and their suitability for the ensiling process and, consequently, for becoming food for farm animals. The research results were presented in a transparent way against the background of available literature. At the end of the work, conclusions are presented, which rightly note certain limitations in translating laboratory research into real conditions. Nevertheless, I consider the work to be a valuable contribution to issues related to the circular economy. I only have a few comments that should be taken into account before publication:
- There is no need to separate subsection 2.5.1 if it is the only one in section 2.5.
Answer: Thank you for your suggestion, we have revised accordingly.
- Please specify how much the mini silos weighed?
Answer: We appreciate your attention to detail in requesting clarification on the weight of the mini silos. The information on the weight of the silos and the weight of the food waste ensiled has been incorporated into the Preparation of mini- silos and the Fermentation characteristics sections.
- The citation of (CFIA, 2015) in section 3.1 is inappropriate. It should be [21].
Answer: Thank you for bringing to our attention the inappropriate citation of (CFIA, 2015) in section 3.1. We have rectified this error and replaced it with the appropriate reference [22].
- In chapter 3.1 there is no description of the studied food wastes (FW). I suggest expanding this chapter.
Answer: Good idea. We have provided a more comprehensive and detailed account of the food waste enriching the content of 3.1.
- The authors did not mention anything about how quickly food waste should be ensiled so that it can be used as food for animals.
Answer: Another good suggestion – this was added to the Conclusions.
Reviewer 3 Report
Comments and Suggestions for Authors
The topic of this manuscript is important – to be able to utilize food waste more efficiently. There are many issues to be solved to make this approach practically feasible, but all of those naturally do not need to be solved in a single manuscript. Maybe challenges related to logistics and technologies in being able to efficiently ensile continuous but rather small amount of material could be mentioned as points needing development. I also started wondering if mechanical separation of liquid and solid such as in Green biorefineries (see e.g., Gaffey, J., Rajauria, G., McMahon, H., Ravindran, R., Dominguez, C., Ambye-Jensen, M., Souza, M.F., Meers, E., Macias Aragonés, M., Skunca, D., & Sanders, J.P.M. 2023. Green Biorefinery systems for the production of climate-smart sustainable products from grasses, legumes and green crop residues. Biotechnology Advances, 66, 108168.) could be one way forward in more efficient utilization of food waste. The challenge in that in that case, two streams, liquid with soluble nutrients and solid with higher fibre content will be produced, which while solving some issues, may generate new challenges in utilization. This approach could fit for the moist vegetables / fruit, while bakery by-products might ne more efficiently utilized as such? This topic just came to my mind, and there is no need to include it in the manuscript.
The design of the experiments was clear, although quite simple. The fact that DM could not be efficiently increased was unfortunate, but as such does not invalidate the experiment. What I was missing was the composition of the material at D0. Could you please add that in the manuscript?
I have some minor comments to develop the manuscript further. It was difficult to give feedback as there was no line numbers in the manuscript. That is why I have included some notes in the pdf document, and I hope it will fine for you to look them up from there.
Results were mostly well presented but please double check: When you present numbers in the scientific text, generally 3 meaningful digits are enough (e.g., 206, 14.9, 0.765). In the number describing the variability, one more digit is used. Please check this throughout the manuscript including text and Tables.
The manufacturers were reported inconsistently in several places. Please check them thoroughly so that name of the product, producer, city, state and country are included.

Comments on the Quality of English Language
Mostly fine.
Author Response
Response to Reviewer #3
Excellent idea for green biorefineries. To most efficiently use food waste, some kind of central processing and distribution system will be required. We are a long way away from that in Canada, though, hence our simple study.
The material at D0 is described on L92-109 and with additional information in Table 1. Additional clarification identifying D0 values have been included in Table 1 in green highlights. Within that overall list of ingredients, there was variation within batches. We initially tried to weigh each constituent before processing but gave up when we were unable to recognize some of the fruits and vegetables. Some Google searching was required after the fact for unfamiliar items. For example, some really spiky vegetables turned out to be Chinese cucumbers. These were lumped in with cucumbers but would have a much different composition than an English cucumber. After some discussion about how cucumbers should be classified - fruits or vegetables, we pressed on with processing without getting too much into food semantics. We recognize that our D0 material is not well described but the diversity did not interfere with preparation of the silage, although it is reflected in ranges in fat content and minerals.
PDF changes have been implemented and are highlighted in green along with other changes in response to reviewer 3 comments. Digits reported have been adjusted as suggested where appropriate. I should have caught this before it went out. Table 3 digits have not been changed as these reflect the limits of detection for these mycotoxins.
Manufacturer reporting has been standardized.